# Role of Pre-Farrow Natural Planned Exposure of Gilts in Shaping the Passive Antibody Response to Rotavirus A in Piglets

**DOI:** 10.3390/vaccines11121866

**Published:** 2023-12-18

**Authors:** Deepak Kumar, Amanda V. Anderson Reever, Jeremy S. Pittman, Nora L. Springer, Kylynn Mallen, Gleyder Roman-Sosa, Neha Sangewar, Mary C. Casey-Moore, Michael D. Bowen, Waithaka Mwangi, Douglas G. Marthaler

**Affiliations:** 1Department of Diagnostic Medicine/Pathobiology, College of Veterinary Medicine, Kansas State University, Manhattan, KS 66506, USAnsangewar@ksu.edu (N.S.); wmwangi@vet.k-state.edu (W.M.); 2Department of Veterinary Diagnostic and Production Animal Medicine, College of Veterinary Medicine, Iowa State University, Ames, IA 50011, USA; amanda_reever@suidaehp.com; 3Smithfield Foods, 434 E Main St., Waverly, VA 23890, USA; jpittman@smithfield.com; 4Clinical Pathology, Biomedical and Diagnostic Sciences, College of Veterinary Medicine, University of Tennessee, Knoxville, TN 37996, USA; nspringer@utk.edu; 5Institute of Virology, University of Veterinary Medicine Hannover, Foundation, 30559 Hannover, Germany; gleyder.roman.sosa@tiho-hannover.de; 6Viral Gastroenteritis Branch, Division of Viral Diseases, National Center for Immunization and Respiratory Diseases (NCIRD), Centers for Disease Control and Prevention, 1600 Clifton Rd NE, Atlanta, GA 30329, USA; quc2@cdc.gov (M.C.C.-M.); mkb6@cdc.gov (M.D.B.); 7Science and Technology, Indical Inc., Orlando, FL 32804, USA; marth027@umn.edu

**Keywords:** antibody, ELISA, natural planned exposure, rotavirus A, sequencing, swine

## Abstract

Natural planned exposure (NPE) remains one of the most common methods in swine herds to boost lactogenic immunity against rotaviruses. However, the efficacy of NPE protocols in generating lactogenic immunity has not been investigated before. A longitudinal study was conducted to investigate the dynamics of genotype-specific antibody responses to different doses (3, 2 and 1) of Rotavirus A (RVA) NPE (genotypes G4, G5, P[7] and P[23]) in gilts and the transfer of lactogenic immunity to their piglets. Group 1 gilts received three doses of NPE at 5, 4 and 3 weeks pre-farrow (WPF), group 2 received two doses at 5 and 3 WPF, group 3 received one dose at 5 WPF, and group 4 received no NPE (control group). VP7 (G4 and G5) and truncated VP4* (P[7] and P[23]) antigens of RVA were expressed in mammalian and bacterial expression systems, respectively, and used to optimize indirect ELISAs to determine antibody levels against RVA in gilts and piglets. In day-0 colostrum samples, group 1 had significantly higher IgG titers compared to the control group for all four antigens, and either significantly or numerically higher IgG titers than groups 2 and 3. Group 1 also had significantly higher colostrum IgA levels than the control group for all antigens (except G4), and either significantly or numerically higher IgA levels compared to groups 2 and 3. In piglet serum, group 1 piglets had higher IgG titers for all four antigens at day 0 than the other groups. Importantly, RVA NPE stimulated antibodies in all groups regardless of the treatment doses and prevented G4, G5, P[7] and P[23] RVA fecal shedding prior to weaning in piglets in the absence of viral challenge. The G11 and P[34] RVA genotypes detected from pre-weaning piglets differed at multiple amino acid positions with parent NPE strains. In conclusion, the results of this study suggest that the group 1 NPE regimen (three doses of NPE) resulted in the highest anti-RVA antibody (IgG and IgA) levels in the colostrum/milk, and the highest IgG levels in piglet serum.

## 1. Introduction

Rotaviruses (RVs) are double-stranded RNA viruses belonging to the Rotavirus genus in the Reoviridae family. The RV genome is approximately 18,522 bp in size and consists of 11 segments of dsRNA encoding six structural proteins (VP1-VP4, VP6 and VP7) and five/six non-structural proteins (NSP 1- NSP5/6) [1]. RV species are classified based on sequencing of the VP6 gene [2,3], and ten RV species, A through J (RVA-RVJ), have been classified [4]. A binary classification system of G (VP7) and P types (VP4) is used to address the vast rotavirus diversity within a species. A complete genome classification system was developed based on the nucleotide sequencing of all 11 RV segments with nucleotide percent identity cut-off values set for each segment, where the VP7-VP4-VP6-VP1-VP2-VP3-NSP1-NSP2-NSP3-NSP4-NSP5/6 RV genes are designated as Gx-P[x]-Ix-Rx-Cx-Mx-Ax-Nx-Tx-Ex-Hx [5,6].

While five RV species (RVA, RVB, RVC, RVE and RVH) have been identified in swine, RVA strains have been considered the most pathogenic and epidemiologically diverse of all porcine RV groups, with detection most common in post-weaning piglets [7]. Prevalence rates ranging from 9.4% to 81.1% have been reported in the US swine population [7,8,9,10,11]. The genotypes G5 (71.43%) and P[7] (77.22%) constitute the most prevalent RVA genotypes circulating in swine herds in the US based on the limited sequencing data from the US [12].

Passive, antibody-based immunity from gilt or sow colostrum and milk is essential to protect piglets from RV infections, since the in-utero transfer of immunoglobulins (Igs) does not occur in swine due to an epitheliochorial placenta. IgA and IgG produced in the sow travels to the mammary glands and is transferred through the colostrum and milk to piglets, where RVs are locally neutralized in the gut [13,14,15,16]. In particular, secretory IgA (sIgA) antibodies play a major role in preventing RV infection at the gut mucosal level to neutralize RV infections [17]. The IgA plasmablasts from the sow gut and IgG from serum transit into the mammary glands, which secrete these immunoglobulins in the colostrum and milk [14,18]. The outer capsid proteins VP7 and VP4 of RVs are targets for the humoral immune response and independently elicit neutralizing and protective antibody responses [19,20,21]. Through the action of trypsin in the gut, VP4 is cleaved into VP8* and VP5* [22]. Both VP8* and VP5* stimulate neutralizing antibodies, and most of the recognized neutralizing epitopes have been mapped to VP8* and the antigen domain of VP5* [23,24].

Although a modified live RVA vaccine (ProSystem RCE, Merck Animal Health) is available that contains G5, G9, P[6] and P[7] genotypes, natural planned exposure (NPE) prior to and during pregnancy is the most widely used method of stimulating lactogenic immunity against RVs in the US swine industry [25]. The NPE method includes the identification of RV-positive fecal samples on a farm and the creation of a “master seed” of RV-infected material. Colostrum-deprived piglets are then inoculated with the RV-infected material and euthanized at 18–24 h, and intestinal material is collected and used to create farm-specific NPE stock [26]. The NPE is fed to pregnant gilts and sows to boost antibody production against specific circulating RVs. However, the efficacy of NPE protocols in providing lactogenic immunity to piglets and shaping the genetic changes in RV strains in piglet populations has not been previously investigated.

In view of the above knowledge gaps, a longitudinal study on a commercial swine farm was designed to test different NPE dosing strategies in gilts for providing lactogenic immunity to their piglets.

## 2. Materials and Methods

### 2.1. Study Design, NPE Material and Sample Collection

This study was conducted on an 1800-head commercial, breed-to-wean, gilt-only farm in the USA. Pregnant gilts were randomly allocated into 4 groups. Group 1 received 3 doses of NPE at 5, 4 and 3 weeks pre-farrow (WPF), group 2 received 2 doses of NPE at 5 and 3 WPF, group 3 received one dose of NPE at 5 WPF and group 4 received no NPE (control group) (Figure 1). Each treatment group initially contained 12 gilts, resulting in 12 piglet litters for each group. Post-farrowing, 2 litters were excluded due to savaging and agalactia. Forty-six litters (group 1 = 12, group 2 = 12, group 3 = 11, group 4 = 11) were evaluated for RVA antibody titers and fecal shedding [26]. Depending on the gilt farrow date, piglets were weaned between 19 and 25 days of age (after the week 3 sampling timepoint) and moved to a separate nursery barn prior to the week 4 sampling timepoint. The piglets continued to be separated by group in the nursery barn.

NPE material on the farm was prepared using the master seed method [25]. In this method, on-farm RV isolates were identified through sampling of pigs with clinical diarrhea and testing via quantitative reverse transcription polymerase chain reaction (qRT-PCR) [25]. Several RV-positive samples free of other harmful pathogens were selected for inoculation in colostrum-deprived piglets. Only RVA and RVC were included in the master seed NPE material due to their more significant production impact. Results of gilts’ and piglet’s antibody responses to RVC in the NPE material were reported by us previously [27]. Colostrum-deprived piglets were obtained by manually catching piglets as they were born, and they were inoculated with the RV-positive material. Piglets were euthanized 18–24 h post-inoculation using carbon dioxide (CO_2_) gas asphyxiation, and their intestines and intestinal contents were harvested, frozen and saved as on-farm NPE stock to be used over the next several months. Diagnostic testing ensured the stock was positive for RV and negative for other relevant pathogens. This method allowed us to have increased confidence that live rotavirus was present in the NPE material while being free of other swine pathogens of concern. VP4 and VP7 gene sequencing was performed on samples from the farm and from the NPE stock to ensure the isolates matched.

To prepare individual doses, 40 mL of the intestinal content material (master seed) was mixed with approximately 14 L of water and enough feed to generate 100 doses (cups) of gruel mixture. Each dose of NPE contained approximately 237 mL of intestinal content material and each gilt received 1 dose of NPE gruel, administered 5 h after daily feeding. A small amount of dry feed was used to bring gilts to the feeder space. Once positioned in their feeding headstalls, 1 dose of NPE was measured and placed into each feeder space. Complete consumption of the NPE dose by all gilts was ensured. A sample of each NPE material was collected for real-time PCR testing and sequencing.

Blood samples from gilts were collected at −5, −3, 0 (farrowing) and 3 weeks. Five piglets from each litter were selected for blood sample collection using BD Vacutainer™ Venous Blood Collection Tubes (Fisher Scientific, Hampton, NH, USA) throughout the study. To assess lactogenic immunity, colostrum was collected at birth and milk was collected 1–3 weeks post-farrowing. Blood samples from 5 piglets per litter were collected at 0 (farrowing), 1, 2, 3, 4, 5 and 6 weeks for a total of 7 blood samples per piglet (Figure 1). Day-0 (at farrowing) blood samples from the piglets were collected within 24 h of colostrum ingestion. No intra-litter movement of pigs was allowed.

### 2.2. Generation of Rotavirus A VP7 and VP4* Expression Constructs

NGS identified G (G4 and G5 VP7) and P (P[7] and P[23] VP4) genotypes in the NPE material. G4 and G5 VP7 constructs were prepared for expression in the mammalian Expi293^TM^ Expression System (Gibco, Waltham, MA, USA). Truncated VP4* gene constructs (aa26-476) of P[7] and P[23] genotypes were generated for bacterial expression. Full length VP7 sequences of G4 and G5 genotypes were modified to add in-frame 8-His tag and Streptavidin tags at the N and C terminals, respectively to track the protein expression and affinity purification of recombinant proteins. Gene sequences were codon-optimized for mammalian expression. A Kozak sequence was also added at the N-terminal to facilitate enhanced protein expression. Linker sequences were added just preceding each affinity tag. A CD5 secretory signal was fused at the N-terminal for efficient secretion of the recombinant protein into the culture media. The dual-tagged synthetic rotavirus VP7 genes were subcloned into a pcDNA3.1 (+) mammalian expression vector (Invitrogen^TM^, Waltham, MA, USA). Truncated VP4* (aa26-476) of the P[7] and P[23] genotypes was cloned into a pET-24a (+) vector with a linker followed by an 8-his tag at the C-terminus. Codon optimization, gene synthesis, cloning into pcDNA3.1 (+) and pET-24a (+) vectors and gene sequence validation were outsourced to Genscript.

### 2.3. Recombinant Protein Expression

G4 and G5 VP7 pcDNA3.1 (+) plasmid constructs were transformed into DH5α competent cells. Positive clones for each construct were identified via PCR screening and used for recombinant protein expression in the mammalian Expi293TM Expression System (Gibco) as per the manufacturer’s protocol. Expi293 cell suspension cultures were transfected with pcDNA3.1 (+) constructs expressing the G4 or G5 VP7. To check the efficiency of protein expression, 300 µL transfected Expi293 cells were plated in a 12-well plate and incubated at 37 °C for 2–3 h. Cells were fixed with ice-cold methanol and used for immunocytometric analysis using a 6x-His Tag monoclonal antibody (1:2000) (MA1–21315, Invitrogen) and goat anti-mouse-IgG antibody labeled with alkaline phosphatase (1:5000) (115-055-146, Jackson Immunoresearch Laboratories, West Grove, PA, USA). To determine whether the protein was secreted or in the cell cytosol, culture media (100 µL) and a small cell pellet were collected for both proteins and an indirect ELISA was performed using the 6x-His Tag monoclonal antibody (1:2500) (MA1–21315, Invitrogen) and goat anti-mouse HRP-conjugated secondary antibody (1:10,000) (31430, Invitrogen). Fractions showing presence of His-tagged proteins were subsequently used for protein purification.

For bacterial expression, pET-24a (+) vectors encoding P[7] and P[23] VP4* (aa26-476) were individually transformed into Rosetta competent cells (70-954-4, Fisher Scientific) and grown overnight on LB agar plates with 30 µg/mL kanamycin at 37 °C. Nest day individual colonies were picked p and amplified overnight in 20 mL of LB broth containing kanamycin at 37 °C with shaking. The overnight culture was added to 1 L of LB broth with kanamycin (30 µg/mL), grown at 37 °C with shaking until it reached an OD600 of approximately 1. Cultures were induced with IPTG added to a final concentration of 0.5 mM for 16 h at 16 °C with shaking. Various time/temperature and IPTG combinations were tested to optimize the production of soluble protein. After its expression, bacterial cultures were centrifuged, and the resulting cell pellets were used for protein purification.

### 2.4. Protein Purification and Validation

Recombinant proteins were purified via immobilized metal affinity chromatography (IMAC) using TALON metal affinity resin (635501, Takara Bio, San Jose, CA, USA) following a hybrid batch/gravity procedure as per the manufacturer’s instructions with modifications. Since G5 VP7 protein was efficiently secreted into the culture media, only Expi293 cell culture supernatant was used to purify G5. G4 VP7 protein was localized inside the cell pellet with less secretion into the culture media, and hence, both the culture supernatant and cell pellet were used to purify G4. Bacterial lysates of P[7] and P[23] VP4* were used for purification. The filtered (0.45 µm) culture supernatant of G5 was directly added to TALON resin. Cell pellets of G4, P[7] and P[23] were resuspended in lysis buffer (NaPO_3_ 50 mM, NaCl 300 mM, imidazole 10 mM, glycerol 10%, pH7.0), homogenized using ultra sonication (30% amplitude, 10 s on and 30 s off, 10 cycles on ice) and centrifuged at 15,000× *g* for 20 min at 4 °C. Protease inhibitor was added to the cell lysates to prevent protein degradation. The supernatant containing soluble protein was filtered (0.45 µm), added to TALON resin and rotated at 4 °C on a rocking platform for 1.5 h to allow for protein binding. The suspension was then centrifuged (700× *g* for 5 min) and the supernatant was discarded. The resin pellet was washed twice with a wash buffer (NaPO_3_ 50 mM, NaCl 300 mM, imidazole 20 mM, glycerol 5%, pH 7.4). The washed resin was transferred to a gravity-flow column and washed again once on the column. His-tagged proteins were eluted using an elution buffer (NaPO_3_ 50 mM, NaCl 300 mM, imidazole 150 mM, glycerol 5%, pH 7.4) in multiple 1.5 mL fractions. Pure protein fractions confirmed by SDS-PAGE were pooled and concentrated using 10 K protein concentrators. Concentrated proteins were quantified using a BCA assay and stored at −80 °C until further use. The affinity-purified proteins were quality-control validated via SDS-PAGE and Western blotting. The expressed proteins were resolved in NuPAGE^®^ Bis-Tris gel (NP0322, Invitrogen^TM^) via denaturing electrophoresis, and stained with AcquaStain (AS001000, Bulldog-Bio, Rochester, NY, USA) for visualization of the protein bands. The proteins were resolved on a gel as above and transferred to an Immun-Blot PVDF membrane (1620177, BioRad, Hercules, CA, USA) via electrophoresis for Western blotting. After transfer, the blot was incubated in blocking buffer (5% non-fat dry milk in 1× PBST) at 4 °C for 1 h, and then probed for 1 h with anti-His monoclonal antibody (1:2000) (MA1–21315, Invitrogen) at room temperature. Following 3 washes with 1× PBST, the blot was incubated with goat anti-mouse HRP-conjugated secondary antibody (31430, Invitrogen) diluted 1:5000 in blocking buffer. Pierce DAB substrate (34002, Thermofisher Scientific, Waltham, MA, USA) was used for chromogenic detection of protein bands.

### 2.5. Development of Recombinant Protein ELISAs to Quantitate RVA Antibodies

Indirect ELISAs were individually optimized to detect genotype-specific RVA IgA and IgG antibodies in porcine serum and colostrum/milk. A checkerboard titration method was used to determine the optimal coating protein concentration for each protein and secondary antibody concentration. Different concentrations (25 ng, 50 ng, 100 ng, 150 ng, 200 ng, 300 ng and 400 ng) of individual proteins were diluted in ELISA carbonate buffer (0.05 M carbonate–bicarbonate, pH 9.6) and coated on 96-well immunoassay plates (2 HB plates, 3355, Thermo Fisher Scientific). Plates were incubated overnight at 4 °C and then washed 4 times using 1× PBST containing 0.05% Tween-20 using an automatic plate washer. Plates were blocked using 5% non-fat dry milk (NFDM) prepared in 1× PBST containing 0.05% Tween-20 at room temperature for 1 h and subsequently washed 4 times. Five RVA-positive serum samples were randomly selected from the sample inventory and diluted 1:200 in 5% NFDM. Diluted serum was added (100 µL) in duplicate to the wells of washed immunoassay plates containing different concentrations of coated proteins. Serum from gnotobiotic piglets was used as the negative control. The plates were incubated at 37 °C for 1 h and washed 4 times using a wash buffer. HRP-conjugated anti-porcine IgG (1:10,000 in 5% NFDM, 100 µL) (ab112748, Abcam, Cambridge, UK) or anti-porcine IgA (1:3000 in 5% NFDM, 100 µL) (ab112746, Abcam) was added to each well and incubated at 37 °C for 1 h. Plates were again washed 4 times with wash buffer, and 100 µL of ABTS substrate was added to each well. Plates were covered and incubated at room temperature for 20 min. The reaction was stopped with 1× ABTS peroxidase stop solution (100 µL). The plates were read using an ELISA microplate reader (Epoch) at 410 nm. The ELISA antibody titer was expressed as the reciprocal of the highest dilution that had a A410 value greater than twice the mean of the negative control wells.

### 2.6. Anti-RVA Antibody Endpoint Titer Determination

Blood and milk samples were processed as described previously [27]. The ELISA protocol detailed in Section 2.5 was used to quantify antibodies against RVA in porcine serum and colostrum/milk samples. To determine the endpoint titer of RVA IgA and IgG antibodies, serum and colostrum/milk samples were serially diluted (1:200, 1:400, 1:800, 1:1600, 1:3200, 1:6400, 1:12,800 and 1:25,600) in 5% NFDM prepared in 1× PBST and added (100 µL) in duplicates to the wells of overnight protein-coated, blocked and washed immunoassay plates. Care was taken to invert the original sample tubes 2–3 times before preparing the dilutions. Washing, time and temperature of incubation, and substrate conditions were the same as detailed in Section 2.5 above. HRP-conjugated anti-porcine IgG and IgA diluted 1:10,000 and 1:3000, respectively, in 5% NFDM 1× PBST were used (Abcam, Cambridge, UK). Each ELISA plate included serially diluted positive and negative controls to control plate-to-plate variation. Since true positive controls (monospecific antiserum against each antigen) were not available, a few high-titer serum samples were pooled and used as positive controls in each plate throughout the ELISA testing to maintain uniformity. Serum from gnotobiotic piglets was used as the negative control. The endpoint titer was expressed as the reciprocal of the highest dilution that had a A410 value greater than twice the mean of the negative control wells.

### 2.7. Next-Generation Sequencing of RV Strains in NPE Material and Piglet Feces

Whole-genome sequencing (WGS) of RV strains in the NPE material was conducted at the Molecular NGS laboratory at Kansas State Veterinary Diagnostic Laboratory (KSVDL), Kansas State University. Piglet fecal samples were chosen for sequencing from week 0–3 samples to assess viruses shed in the presence of lactogenic immunity (Table 1). Litters from which RVA was detected for multiple weeks in a row with Ct values less than 26 were selected for sequencing. Four more piglet fecal samples from week 4 (nursery) were also included for sequencing to determine RVA genotypes circulating outside the lactogenic immune pressure (Table 1). WGS of piglet fecal samples was conducted by the Rotavirus Surveillance and Molecular Epidemiology Team at the Centers for Disease Control and Prevention (CDC), Atlanta, Georgia. Viral dsRNA extraction, cDNA library synthesis, NGS techniques and analysis were performed as previously described [28].

### 2.8. Statistical Analysis

The significance of the differences between the treatment and the control groups was determined via a two-way analysis of variance (ANOVA). All statistical analyses were performed using GraphPad Prism 7 (Version 7.04, GraphPad Software, Inc., La Jolla, CA, USA) and a significance level of *p* < 0.05 was used for all analyses.

## 3. Results

### 3.1. Expression of Recombinant Proteins and Optimization of ELISAs

The ELISA using the anti-His monoclonal antibody suggested efficient G5 protein secretion into the Expi293 culture media; however, the G4 protein was localized inside the cell pellet with less secretion into the media (results not shown). The immunocytometric staining of HEK-293A cells transfected with pcDNA3.1 (+) plasmids encoding G4 or G5 VP7 genes and probed with anti-His monoclonal antibody gave positive staining and confirmed recombinant protein expression (Figure 2A,B). No staining was observed in the mock transfected HEK293 cells (Figure 2C). An estimated 37 kDa and 55 kDa bands corresponding to the expected molecular weights of recombinant VP7 and truncated VP4* proteins, respectively, were detected on SDS-PAGE (Figure 2D–F). The authenticity of the antigens was validated via Western blot using anti-His monoclonal antibodies (Figure 2G). The protein concentrations of 50 ng (G4 VP7), 100 ng (G5 VP7 and P[7] VP4*) and 150 ng (P[23] VP4*) resulted in optimal OD value readouts for indirect ELISA standardization. Blocking the immunoassay plates with 5% NFDM prepared in 1× PBST with 0.05% Tween-20 and performing four washings after each incubation step resulted in minimal background levels. The optimal incubation temperature and time combination for the samples (serum/colostrum/milk) and secondary antibodies was 37 °C for 1 h. Respective concentrations of 1:3000 and 1:10,000 for peroxidase-conjugated IgA and IgG were found to produce the best OD readouts.

### 3.2. Antibody Response to RVA NPE

#### 3.2.1. Gilt Serum

Out of 46 gilts, fecal samples of only 2 gilts from the control group were found to be positive for RVA via qRT-PCR at 5 weeks before farrowing (−5 W) [26]. Since RVA is prevalent in swine herds, the study gilts had likely experienced RVA infection prior to their enrollment in the study, which resulted in varied levels of IgA and IgG antibodies before the administration of the first NPE dose at 5 weeks pre-farrow (−5 W) (Figure 3A–H). The geometric mean titer (GMT) IgG levels at −5 W were G5 (GMT 1067–1932), G4 (GMT 1243.53–1940.94), P[7] (GMT 5079.68–10,159.37) and P[23] (GMT 4381.12–6816.26). The GMT IgA levels at −5 W were G5 (GMT 400–1029.33), G4 (GMT 548–852.03), P[7] (GMT 1704.07–2539.84) and P[23] (GMT 1243.53–2334.17). At −5 W, the control group IgA and IgG levels were higher compared to those of the treatment groups, except for the P[7] IgG levels, where the group 2 IgG levels were slightly higher than those of the control group (Figure 3A–D). Two doses of NPE in group 1 (5 and 4 weeks before farrowing) and one dose each in groups 2 and 3 (5 weeks pre-farrow) resulted in increased G5 and G4 IgG levels at −3 W (3 weeks-pre-farrow) compared to the control group, which decreased at −3 W in the absence of NPE (Figure 3A–D). For P[7], only group 1 showed elevated IgG levels at −3 W. In contrast, P[23] showed increased IgG levels at −3 W for both groups 1 and 2. Gilt serum IgG levels dropped sharply in all treatment groups at farrowing. Serum IgG levels gradually increased after farrowing until 3 weeks post-farrowing (+3 W, weaning).

Serum IgA levels for treatment groups 1, 2 and 3 increased at −3 W after administration of the respective NPE doses (Figure 3E,F). The control group serum IgA levels decreased at −3 W for G4 and G5 in the absence of the first NPE dose. Interestingly, an increase in the serum IgA levels of the control group was observed for P[7] and P[23] at −3 W (Figure 3G,H). Like IgG levels, IgA levels also increased post-farrowing until weaning.

#### 3.2.2. Colostrum and Milk

Colostrum and milk samples were collected at farrowing (day 0), and then, at weekly intervals until weaning (days 7, 14 and 21). Day-0 colostrum antibody levels in RVA genotypes have been previously reported by us [27]. At day 0, treatment group 1 (3 NPE) had significantly higher G5 and G4 IgG Ab titers compared to group 2 (2 NPE), group 3 (1NPE) and the control group (Figure 4A,B). The P[7] and P[23] IgG Ab titers at day 0 were significantly higher in treatment group 1 compared to the control group (Figure 4C,D). Overall, at day 0, treatment group 1 had significantly higher colostrum IgG titers for all antigens compared to the control group, and either significantly or numerically higher IgG titers than groups 2 and 3 (Figure 4A–D). In addition, the colostrum IgG levels for all antigens were highest on day 0, which rapidly declined and reached the baseline at day 7 in milk and remained so during the subsequent samplings (Figure 4A–D).

The G5 IgA Ab titers were significantly higher in group 1 (3 NPE) compared to group 2 and the control group, while the G4 IgA Ab titers were not significantly different for any treatment group (Figure 4E,F, Table 2). The P[7] and P[23] VP4* IgA Ab titers were significantly higher in group 1 compared to other treatment groups and the control group (Figure 4G,H, Table 2). Overall, at day 0, colostrum IgA levels for all antigens (except G4) were significantly higher in group 1 compared to the control group, and either significantly or numerically higher compared to groups 2 and 3 (Figure 4E–H, Table 2). G5 VP7 IgA titers declined at day 7, and then, gradually increased until they reached the same Ab titers as on day 0 for all groups except group 1 (Figure 4E). The P[7] and P[23] IgA titers never reached the same values as on day 0 (Figure 4G,H). Lastly, the VP4*-specific IgG and IgA titers were at least five times higher than the VP7-specific IgG and IgA titers (Figure 4A–H).

#### 3.2.3. Piglet Serum

Piglet serum samples were collected at birth, and then, at weekly intervals until 6 weeks of age (day 42). At birth (day 0), piglets born to treatment group 1 gilts had significantly high IgG Ab titers against G5, P[7] and P[23] compared to group 2, group 3 and the control group piglets (Figure 5A,C,D). Group 1 piglets had significantly higher G4 IgG titers compared to group 3 and the control group at day 0 (Figure 5B). P[7] IgG titers for all groups declined post-birth, reaching the baseline at day 42 (Figure 5C). P[23] IgG Ab titers for treatment groups 1, 2 and 3 declined at day 7, with groups 2 and 3 showing a slight increase at days 14 and 21. P[23] VP4* IgG titers increased slightly at day 42 (Figure 5D). Interestingly, P7 IgG levels in all three treatment groups on day 7 did not decrease as sharply as for the other proteins (Figure 5C). Significant differences were observed in IgG levels among the different groups at multiple time points for all four antigens.

Group 3 serum samples had higher IgA Ab titers at day 0 and 7 for all four antigens compared to groups 1 and 2 and the control group, although the levels were not significantly different than those in group 1 piglets (Figure 5E–H, Table 2). Overall, for all antigens, the serum IgA levels of all four groups were highest at day 0, rapidly declined at day 7, and reached the baseline at weaning (day 21). Similar to serum IgG titers, IgA levels differed significantly among different groups at days 0 and 7.

### 3.3. Effects of NPE Administration in Gilts on Piglet Fecal RVA Shedding

The Real-time PCR of NPE material revealed RVA Ct-values of 24.43, 22.46 and 24.15 for NPE 1 (5WPF), 2 (4WPF) and 3 (3 WPF), respectively. Gilt and piglet RVA fecal shedding results have been reported previously [26]. All piglets’ fecal swabs collected within 24 h of farrowing were negative for RVA according to qRT-PCR [26]. Out of 46 litters, a single litter in treatment groups 1 (litter 41049) and 3 (litter 40996) shed RVA for multiple weeks prior to weaning (Appendix A). The serum antibody levels of the two piglets shedding RVA pre-weaning, along with the colostrum IgA levels of the respective gilts, are summarized in Table 3. Piglets belonging to the litter 40996 had overall higher serum IgA levels for P[23], G5 and G4 at 1 week of age compared to the piglets of litter 41049 (fecal RVA Ct-value: 16.09). The RVA-positive pre-weaning samples and four samples at week 4 were selected for NGS to investigate genetic changes in response to immunity (Table 1). A complete RVA genome could only be recovered from the group 3, week 2 sample (litter 40996) while the four samples at week 4 also yielded complete RVA genomes (Table 4). Sequencing revealed an RVA genome constellation of G11-P[34]-I5-R1-C1-M1-A8-N1-T7-E9-H1 from the week 2 sample (litter 40996). The week 4 sample of litter 40996, along with other week 4 samples, yielded a genome constellation of G9-P[23]-I5-R1-C1-M1-A8-N1-T7-E1-H1. The G- and P-type combination (G11P[34]) detected in a pre-weaning sample of litter 40996 was different from the genotypes present (G4, G5, P[7] and P[23]) in the original NPE material fed to the gilts (Table 4).

### 3.4. Sequence Analysis and Antigenic Variation among the RVA Strains

We found that the G11 VP7 sequence (GenBank accession number OQ291257) from piglets in the farrowing room shared 82.77% nucleotide and 89.57% amino acid percent identity with the G5 sequence of the NPE. However, the nucleotide and amino acid percent identities with the G4 NPE sequence were lower—72.58% and 75.46%, respectively. G9 VP7 sequences (GenBank accession numbers OQ326585-OQ326588) from the nursery shared low nucleotide (75.23–78.29%) and amino acid (78.22–83.13%) percent identity with the parent G4 and G5 sequences. To further examine the sequence variations, neutralizing epitopes on the VP7 of five RVA sequences recovered from piglet feces were compared to the parent G4 and G5 sequences. Out of 34 residues present in the neutralizing epitopes, 9 residues were conserved (D95, S103, K143, S190, T192, T209, T210, E216 and K223) (Table 5). The G11 strain expressed the highest number of differences from the G4 strain (*n* = 21) compared to the G5 strain (*n* = 10) of the NPE. All G9 sequences completely differed from the G4 and G5 sequences at 10 amino acid positions (90, 94, 100, 122, 147, 189, 208, 212, 213 and 221). A series of common B-cell epitopes for RVA VP7 spread across multiple RVA genotypes have been proposed recently [13]. These include residue positions 87, 90–92, 94–97, 99, 122, 147 and 210–213. Higher amino acid variability was observed at these positions, with five piglet RVA VP7 sequences differing either from the parent G4 or G5 sequence at multiple residues (Table 5).

For VP4, out of 34 residues spread across the VP8* and VP5* regions, 9 residues completely matched the P[7] and P[23] sequences of the NPE (D100, Q125, N132, G150, N193, Y194, Y385, G392 and R425). The P[34] strain (GenBank accession number OQ291258) recovered from pre-weaning piglets expressed the highest number of differences from the parent P[7] strain (*n* = 22), followed by the P[23] strain (*n* = 17) (Table 6). P[23] sequences (GenBank accession numbers OQ326589–OQ326592) from piglets showed differences at 17 amino acid sites when compared to the parent P[7] strain. However, the P[23] sequence only differed at residue 188 (Y188T) with the parent NPE P[23] strain (Table 6).

## 4. Discussion

This study was designed to investigate the dynamics of antibody responses to different dosing regimens of NPE to gilts and the transfer of immunity to their piglets. To evaluate the effectiveness of RV NPE protocols and differences in antibody responses to different proteins, indirect ELISAs were developed to investigate antibody responses to the RVA G4, G5, P[7] and P[23] genotypes in gilt serum, colostrum/milk and piglet serum samples. We also investigated the association of antibody levels with RVA genotypes and fecal RVA shedding in piglets. Sequence analysis was performed to determine genetic changes in RVA genotypes recovered from piglets in the presence of NPE.

Two doses of NPE in group 1 and one dose of NPE in treatment groups 2 and 3 resulted in increased gilt serum IgA levels at 3 WPF, reflecting the stimulation of active immunity against RVA in gilts. As expected, gilt serum IgG levels in all treatment groups dropped sharply at farrowing due to the transudation of serum immunoglobulins into the swine colostrum [29]. However, serum IgA levels at farrowing did not drop as distinctly as IgG, which could be either due to the increased numbers of IgA-producing cells at sub-mucosal sites raising serum IgA levels through uptake from the lymphatics, the release of IgA locally produced in the mammary glands into gilt serum, or the reduced transportation of serum IgA into exocrine fluids [30]. Like our results, Klobasa and coworkers also found elevated sow serum IgA levels during the last weeks of gestation, in contrast to serum IgG levels, which dropped sharply at farrowing.

Our results indicated that group 1 gilts with three doses of NPE had significantly higher colostrum anti-RV IgG titers for all antigens compared to the control group. For all groups, colostrum IgG levels declined sharply and reached the baseline (dilution 1:200) on day 7. Rapid declines in colostrum IgG levels occurred in parallel with rapid increases in sow serum IgG titers post-farrowing until weaning. Similar to our results, high RV-specific colostrum antibody titers (8–32 fold) have been reported compared to milk collected at 18 days post-farrowing [31]. Overall, three doses of NPE in group 1 resulted in higher colostrum IgG levels at day 0 compared to other groups for all antigens. We observed that three doses of NPE in group 1 also resulted in higher anti-RV IgA levels in colostrum (day 0) compared to the control group for all antigens. In a study comparing the efficacy of maternally derived anti-RV antibodies on piglet protection against RVA, significantly higher RVA-specific IgG levels and anti-RVA virus neutralization titers (1600 versus 340) were reported for immune colostrum (collected from sows immunized with RVA) compared to conventional colostrum (non-immunized) [15]. We also found that colostrum IgA levels were highest at day 0, followed by a decline at day 7, and then, a steady increase at day 14 until day 21, which reflects the increased number of RVA-specific IgA plasmablasts in the mammary gland tissue and continuous supply of secretory-IgA in the colostrum and milk throughout lactation. High pathogen-specific IgA levels in milk have been associated with lower incidence of enteric disease, including RV, in swine [15,18]. In addition, RVA IgA and IgG levels in the colostrum and milk of RVA-exposed field sows decreased gradually over time and provided protection against RVA infection in piglets for the first 1–2 weeks in a virus challenge model [31]. Overall, group 1 gilts had higher colostrum IgA levels (day 0) for all antigens (except G4) compared to the other study groups. High IgA levels in milk from birth until weaning substantiate the role of IgA in providing protection against RV infection prior to weaning.

The maternally derived IgG and IgA levels in piglet serum were highest on day 0, and then, declined thereafter, which could be attributed to the cessation of the absorption of intact immunoglobulins (gut closure) approximately 18–36 h post-birth [32]. The antibodies provided in colostrum and milk play a crucial role in the protection of piglets against rotaviral infection [15]. The ingested maternal Igs remain intact within the digestive tract, probably because of the low proteolytic activity of piglets’ digestive tracts [33]. As expected, IgA levels declined more sharply across sampling points because of their shorter half-life of approximately 6 days as compared to 24 days for IgG [34,35].

For all antigens, piglet serum IgG levels reached low levels on day 28 (a week after weaning). Compared to the other proteins, P[23] IgG titers increased at day 42, probably due to the development of active immunity to the P[23] genotype present in the nursery. In contrast, no increase in day-42 piglet serum IgG titers for G4, G5 and P[7] was detected, which was supported by no detection of these genotypes following the sequencing of selected piglet fecal samples. However, the absence of G4, G5 and P[7] genotypes cannot be confirmed due to the limited number of samples sequenced.

RVA fecal shedding in piglets peaked at day 28, one week after the piglets moved to the nursery (Appendix A), which could be mainly attributed to the loss of lactogenic protection post-weaning. A rapid decline in piglet serum IgA titers was observed starting on day 14. At weaning (day 21), the piglet serum IgA levels were very low and ranged from 200 to 350 for all antigens, while serum IgG levels were higher and ranged from 1156 to 2645 for P-types and 266 to 672 for G-types. Although serum IgA is considered a good indicator of intestinal IgA levels [36,37], studies suggest that serum IgG also provides protection against RV infection [38,39]. Piglet serum IgA levels remained very low until the last day of sample collection (day 42). A comparison of piglet serum IgA titers and piglet fecal RVA shedding points towards a possible high environmental burden of RVA in the nursery, which may have been another reason for high fecal shedding in piglets on day 28 considering the very low serum IgA levels in piglets during days 14–42.

Similar to our findings, low levels of anti-RVA antibodies have been reported in piglet serum at 3 weeks of age [31]. The authors were also able to determine protective levels of anti-RVA antibody titers and reported that piglets shed RVA when antibody titers in the serum fell below 1/1600 at day 21 of age. In our study, a positive correlation was not observed between the levels of anti-RV antibody and protection in individual litters, as the two piglets shedding RVA in the farrowing room had serum IgA levels of 1:200–1:400 (G-types) and 1:1600–1:3200 (P-types) at the time of shedding (1 week of age). It is important to note that we determined protein-based (VP7 and VP4*) endpoint titers in contrast to the whole virus antibody titers by Fu and coworkers, and hence, any comparison of antibody titers between two studies is likely to be ambiguous.

The outer capsid proteins VP7 and VP4 of RVs are targets for the humoral immune response and independently elicit neutralizing and protective antibody responses [20,21,40]. The full-length VP7 and truncated version of VP4* (aa26-476) proteins were generated as expression constructs, as described in previous studies [24,41]. Our IgG and IgA GMTs against VP4* were at least five times higher compared to VP7 protein, which is consistent with previously published reports published. Ishida et al. [42] reported 9–27 times higher serum IgG titers for VP4 (titer = 1350) compared to VP7 (titers 50 and 150) against recombinant baculovirus-expressing EHP VP4 and RRV VP7 in mice. Similarly, another study found that the magnitude of homotypic IgA antibody responses (fold GMT increase) in infant serum to VP4 was higher than VP7 in all study groups [43]. This difference in the magnitude of antibody response between VP4 and VP7 could be due to fewer neutralizing epitopes (NEs) on VP7 (*n* = 4) compared to 9 NEs on VP4. However, the difference in endpoint antibody titers does not necessarily reflect the differences in the ability of these proteins to stimulate a neutralizing antibody response.

It was observed that antibody levels in piglets’ serum for all study groups were able to prevent pre-weaning RVA fecal shedding in a genotype-specific manner (Appendix A) [26]. Although two litters shed RVA prior to weaning, these litters belonged to a G- and P-type combination (G11P[34]) different from the NPE material administered to the gilts, which suggests that passive homotypic immunity was stimulated against RVAs by NPE. This assumption was further strengthened by the detection of the P[23] genotype (similar to NPE) in the piglet feces from treatment groups collected outside the window of lactogenic immunity (post-weaning samples).

Even though the G11 strain shared 89.57% amino acid percent identity with the parental G5 strain, a set of point-mutations at 10 key amino acid sites was observed. Of these, seven sites (91, 96, 99, 122, 211–213) had previously been predicted as shared B-cell epitopes for RVA [44]. These predicted B-cell epitopes (pBCEs), shared across multiple RVA genotypes, have been proposed to be the common structural targets of RVA antibodies regardless of genotype [44]. Several of these pBCEs are also known neutralization escape mutation sites (NEM) for RVAs (Table 5). High amino acid diversity at these pBCEs and NEMs could make them key targets of porcine antibodies and lead to the development of virus immune escape mutants. We hypothesize that multiple point-mutations at shared amino acid sites might have resulted in the development of immune escape mutants, which was confirmed by the detection of the G11P [34] genotype combination in the farrowing room.

One of the limitations of this study was not challenging the newborn piglets with RVA. Using a virus challenge model would have truly estimated the differences in NPE doses in providing passive protection to the piglets. It is likely that piglets in the farrowing room may not have been sufficiently challenged by environmental RVA. No piglet viral challenge and no RVA fecal shedding in the control group prevented the comparison of protective efficacy of different NPE doses. Our results emphasize the need for the routine surveillance and genotypic analysis of RV genotypes circulating on swine farms, including environmental samples, because we do not know whether the RVA strains detected from piglet feces before weaning were already present in the farrowing room environment or resulted from immune escape mutations.

However, it is important to note that this study was conducted on a gilt-only commercial farm to assess the effectiveness of NPE in natural farm conditions. Selecting a gilt-only farm also ensured the most challenging conditions since gilts have been reported to have lower colostrum rotavirus antibody levels than multiparous sows. Most importantly, the absence of fecal RVA shedding in the control group suggests that pre-existing RVA antibody levels in the control group were able to prevent homologous RVA infection in natural farm conditions, and hence, multiple pre-farrow RVA NPE doses might not be required.

## 5. Conclusions

Treatment group one with three doses of pre-farrow NPE resulted in significantly higher IgG and IgA levels in the colostrum and milk, and higher IgG levels in the piglets’ serum. RVA NPE stimulated antibodies in all groups regardless of the treatment dose and prevented G4, G5, P[7] and P[23] RVA infection prior to weaning. RVA is more prevalent and pathogenic compared to other porcine RVs, and every sow typically experiences multiple RVA infections within its lifetime. Therefore, it is highly likely that sows harbor more RVA-specific memory B cells than other non-RVA-specific memory B cells, which, upon repeated exposure to RVA antigens, proliferate and differentiate into RVA-specific antibody producing plasma cells. Control group gilts, with no viral stimulation in the form of “NPE”, prevented RVA fecal shedding in piglets in the farrowing room, and piglets born to group 3 gilts had higher serum IgA levels compared to the other groups. Overall, colostrum/milk and piglet serum antibody levels, and RVA fecal shedding in piglets, suggest that low antibody titers were sufficient to prevent homologous RVA infection in piglets until weaning (day 21). In view of the above observations, we recommend using one dose of “RVA only NPE” 5 weeks prior to farrowing.

## Figures and Tables

**Figure 1 vaccines-11-01866-f001:**
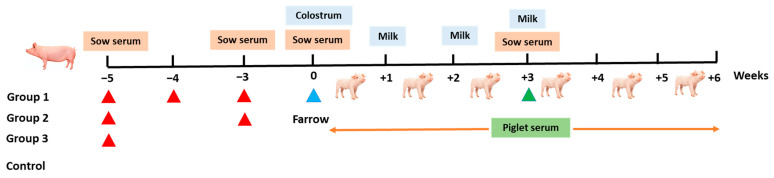
NPE administration and sample collection (serum, colostrum, and milk) schedule. NPE administration to gilts, farrowing and weaning are indicated by red, blue and green triangles. Gilts (N = 12 per group) and five piglets per gilt were sampled individually. Control group gilts did not receive NPE.

**Figure 2 vaccines-11-01866-f002:**
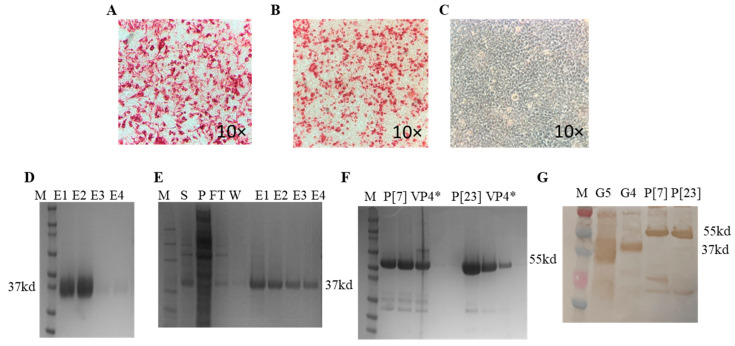
Validation of antigen expression via immunocytometric staining and immunoblotting. (**A**) HEK293A cells transfected with pcDNA3.1 (+) construct encoding G5 VP7 RVA showing positive anti-His tag staining. (**B**) HEK293A cells transfected with pcDNA3.1 (+) construct encoding G4 VP7 RVA showing positive anti-His tag staining. (**C**) Negative control. (**D**) Purified G5 RVA VP7 (37 kd). (**E**) Purified G4 RVA VP7 (37 kd). M—protein marker; S—culture supernatant; P—cell pellet; FT—flow through; W—wash fraction; E1, E2, E3 and E4—sequential G4 protein elutes. (**F**) Purified P[7] and P[23] VP4* (55 kd) protein elutes. (**G**) Western blot confirmation of the affinity-purified G5, G4, P[7] and P[23] antigens using anti-His monoclonal primary and HRP-tagged anti-mouse IgG secondary antibodies.

**Figure 3 vaccines-11-01866-f003:**
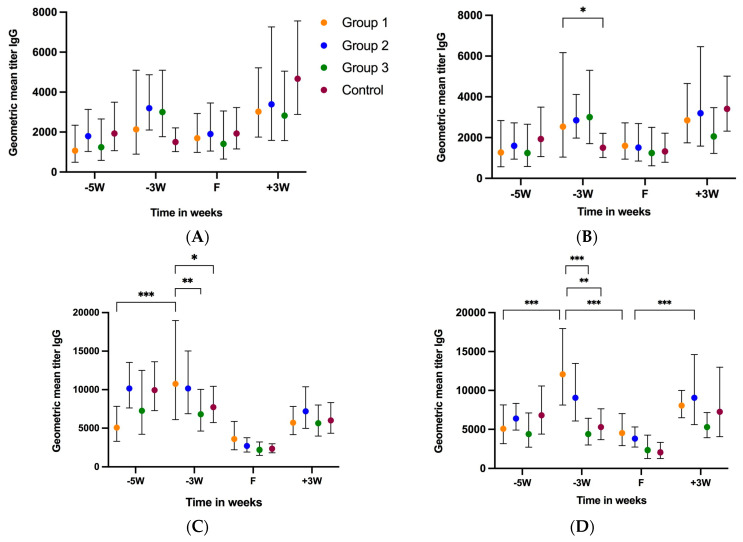
Kinetics of gilt serum antibody levels against RVA. Progression of serum IgG and IgA levels over time in gilts receiving three (group 1), two (group 2), one (group 3) or no (group 4, control) doses of NPE. Gilt serum IgG levels against G5 (**A**), G4 (**B**), P[7] (**C**) and P[23] (**D**) RVA antigens. Gilt serum IgA levels against G5 (**E**), G4 (**F**), P[7] (**G**) and P[23] (**H**) RVA antigens. The horizontal axis represents sample collection time-points (−5 W = 5 weeks pre-farrow; −3 W = 3 weeks pre-farrow; F = at farrowing; +3 W = 3 weeks post-farrow or at weaning). The vertical axis represents geometric mean antibody titers (GMTs). * *p* < 0.033, ** *p* < 0.002, *** *p* < 0.001, mixed effects model.

**Figure 4 vaccines-11-01866-f004:**
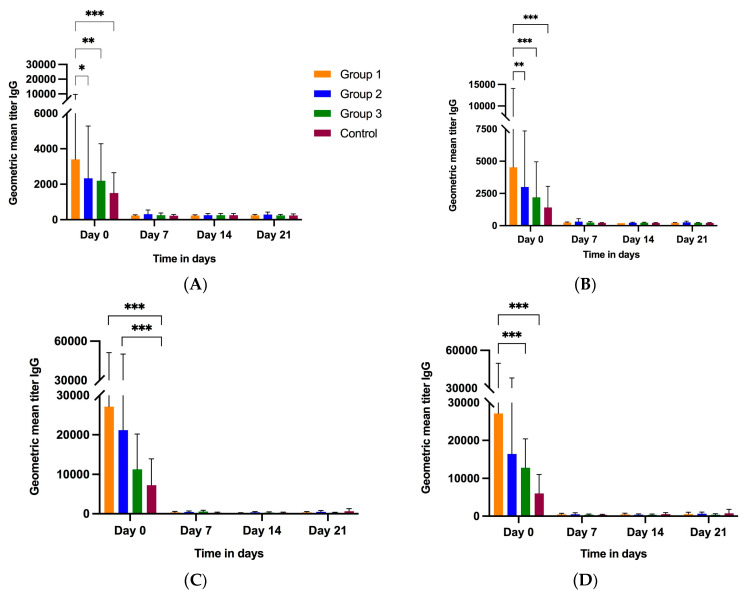
Kinetics of colostrum/milk antibody levels against RVA. Progression of colostrum/milk IgG and IgA levels over time in gilts receiving three (group 1), two (group 2), one (group 3) or no (group 4, control) doses of NPE. Colostrum/milk IgG levels against G5 (**A**), G4 (**B**), P[7] (**C**) and P[23] (**D**) RVA antigens. Colostrum/milk IgA levels against G5 (**E**), G4 (**F**), P[7] (**G**) and P[23] (**H**) RVA antigens. The horizontal axis represents sample collection time-points (−5 W = 5 weeks pre-farrow; −3 W = 3 weeks pre-farrow; F = at farrowing; +3 W = 3 weeks post-farrow or at weaning). The vertical axis represents geometric mean antibody titers (GMTs). * *p* < 0.033, ** *p* < 0.002, *** *p* < 0.001, mixed effects model.

**Figure 5 vaccines-11-01866-f005:**
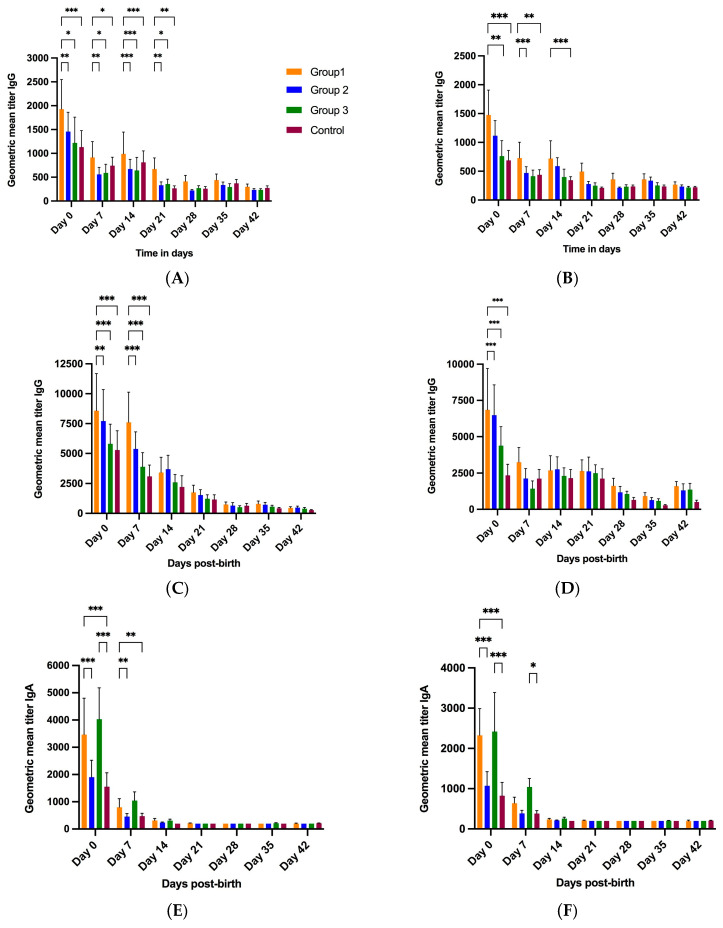
Kinetics of piglet serum antibody response to RVA. Progression of piglet serum IgG and IgA levels over time in piglets born to gilts receiving three (group 1), two (group 2), one (group 3) or no (group 4, control) doses of NPE. Piglet serum IgG levels against G5 (**A**), G4 (**B**), P[7] (**C**) and P[23] (**D**) RVA antigens. Piglet serum IgA levels against G5 (**E**), G4 (**F**), P[7] (**G**) and P[23] (**H**) RVA antigens. The horizontal axis represents sample collection time-points (−5 W = 5 weeks pre-farrow; −3 W = 3 weeks pre-farrow; F = at farrowing; +3 W = 3 weeks post-farrow or at weaning). The vertical axis represents geometric mean antibody titers (GMTs). * *p* < 0.033, ** *p* < 0.002, *** *p* < 0.001, mixed effects model.

**Table 1 vaccines-11-01866-t001:** Piglet fecal samples sequenced for RVA variant detection.

Litter Id.	Group	Week	RVA Ct
41049	1	1	15.42
		2	18.62
40996	3	2	18.48
		3	17.28
41071	1	4	13.07
41045	2	4	11.29
40996	3	4	12.62
41053	4	4	12.15

**Table 2 vaccines-11-01866-t002:** Treatment group-specific colostrum and piglet serum IgA geometric mean titers (GMT) at farrowing.

Group	Colostrum IgA Titer (GMT, Day 0)				Piglet Serum IgA Titer (GMT, Day 0)			
	G4	G5	P[7]	P[23]	G4	G5	P[7]	P[23]
1	3020	4278	24,163	24,163	2170	3466	11,353	8145
2	2193	2821	14,519	10,595	1025	1903	6315	4406
3	3200	3200	15,464	10,595	2216	3966	15,097	10,500
4	1411	1600	3408	3200	893	1748	4981	3820

**Table 3 vaccines-11-01866-t003:** Serum IgA levels of two piglets shedding RVA in the farrowing room, and colostrum IgA levels of respective sows at day 0.

Litter	Antigen	Sow Colostrum IgA (Day 0)	Piglet Serum IgA Levels at Multiple Time Points							
			Day 0 (Group GMT)	Day 0	Day 7	Day 14	Day 21	Day 28	Day 35	Day 42
41049	P[7]	12,800	11,353	25,600	1600	200	200	200	1600	3200
	P[23]	25,600	8145	12,800	1600	400	200	200	1600	1600
	G5	800	3466	3200	200	200	200	200	200	200
	G4	800	2275	1600	200	200	200	200	200	200
Fecal RVA shedding (Ct Value)				-	16.09	20.96	27.13	12.80	26.96	25.13
40996	P[7]	25,600	15,097	12,800	1600	1600	800	200	200	200
	P[23]	25,600	10,500	25,600	3200	800	1600	200	200	200
	G5	800	3966	800	400	200	200	200	200	200
	G4	1600	2340	3200	400	200	200	200	200	200
Fecal RVA shedding (Ct value)				-	28.09	20.09	15.5	14	22.62	18.07

**Table 4 vaccines-11-01866-t004:** Genome constellation of RVA strains detected in piglet feces.

Litter	Group	Week	RVA Ct	Genome Constellation
40996	3	2	18.49	G11-P[34]-I5-R1-C1-M1-A8-N1-T7-E9-H1
		4	12.62	G9-P[23]-I5-R1-C1-M1-A8-N1-T7-E1-H1
41071	1	4	13.07	G9-P[23]-I5-R1-C1-M1-A8-N1-T7-E1-H1
41045	2	4	11.29	G9-P[23]-I5-R1-C1-M1-A8-N1-T7-E1-H1
41053	4	4	12.15	G9-P[23]-I5-R1-C1-M1-A8-N1-T7-E1-H1

**Table 5 vaccines-11-01866-t005:** Antigenic variation in amino acid sequences of VP7 proteins among the RVA strains (G11 and G9) recovered from piglet feces and parent NPE strains (G4 and G5).

	NEM		NEM		NEM		NEM	NEM																					NEM				NEM	NEM
	87	90	91	92	94	95	96	97	99	100	103	119	122	123	124	126	143	144	147	152	189	190	192	208	209	210	211	212	213	214	216	221	223	242
**G4-NPE-Parent**	**T**	**R**	**T**	**O**	**N**	**D**	**N**	**E**	**K**	**D**	**S**	**N**	**S**	**N**	**V**	**E**	**K**	**F**	**G**	**I**	**S**	**S**	**T**	**Q**	**T**	**T**	**N**	**A**	**N**	**T**	**E**	**S**	**K**	**T**
G11-Piglet	N	A	R	E	A	.	D	K	.	.	.	K	T	D	I	S	.	Y	N	M	T	.	.	L	.	.	.	S	A	.	.	A	.	A
G9-Piglet	N	S	.	O	G	.	T	.	.	N	.	K	T	D	I	S	.	Y	T	M	O	.	.	T	.	.	.	T	A	.	.	N	.	N
G9-Piglet	N	S	.	O	G	.	T	.	.	N	.	K	T	D	I	S	.	Y	T	M	O	.	.	T	.	.	.	T	A	.	.	N	.	N
G9-Piglet	N	S	.	O	G	.	T	.	.	N	.	K	T	D	I	S	.	Y	T	M	O	.	.	T	.	.	.	T	A	.	.	N	.	N
G9-Piglet	N	S	.	O	G	.	T	.	.	N	.	K	T	D	I	S	.	Y	T	M	O	.	.	T	.	.	.	T	A	.	.	N	.	N
**G5-NPE-Parent**	**N**	**A**	**T**	**E**	**A**	**D**	**T**	**K**	**T**	**E**	**S**	**K**	**A**	**D**	**I**	**S**	**K**	**Y**	**N**	**M**	**T**	**S**	**T**	**S**	**T**	**T**	**D**	**I**	**N**	**S**	**E**	**A**	**K**	**N**
G11-Piglet	.	.	R	.	.	.	D	.	K	D	.	.	T	.	.	.	.	.	.	.	.	.	.	L	.	.	N	S	A	T	.	.	.	.
G9-Piglet	.	S	.	O	G	.	.	E	K	N	.	.	T	.	.	.	.	.	T	.	O	.	.	T	.	.	N	T	A	T	.	N	.	.
G9-Piglet	.	S	.	O	G	.	.	E	K	N	.	.	T	.	.	.	.	.	T	.	O	.	.	T	.	.	N	T	A	T	.	N	.	.
G9-Piglet	.	S	.	O	G	.	.	E	K	N	.	.	T	.	.	.	.	.	T	.	O	.	.	T	.	.	N	T	A	T	.	N	.	.
G9-Piglet	.	S	.	O	G	.	.	E	K	N	.	.	T	.	.	.	.	.	T	.	O	.	.	T	.	.	N	T	A	T	.	N	.	.

Note: The residues of the Parent NPE strains (G4 and G5) are in bold, while dots represent the same residues compared to the NPE strains listed above. Red numerals indicate predicted B cell epitopes for RVA. NEM in green color indicates known RVA-neutralizing escape mutation sites (NEM).

**Table 6 vaccines-11-01866-t006:** Antigenic variation in amino acid sequences of VP4 proteins among the RVA strain (P[34] and P[23]) recovered from piglet feces and parent NPE strains (P[7] and P[23]).

	**VP8***																	**VP5***																
	87	88	89	100	113	114	115	116	125	131	132	133	135	146	148	150	173	180	183	188	190	192	193	194	195	196	217	385	388	392	393	425	433	458
**P[7]-Parent**	**T**	**V**	**E**	**D**	**Q**	**T**	**T**	**N**	**Q**	**E**	**N**	**T**	**Q**	**T**	**P**	**G**	**R**	**T**	**N**	**Y**	**S**	**T**	**N**	**Y**	**D**	**T**	**T**	**Y**	**A**	**G**	**A**	**R**	**G**	**Q**
P[34]-Piglet	K	N	D	.	I	S	.	T	.	S	.	M	T	Q	S	.	L	E	.	Q	T	S	.	.	S	E	.	.	R	.	K	.	L	D
P[23]-Piglet	S	N	A	.	P	S	E	S	.	.	.	V	T	.	I	.	K	E	.	F	T	.	.	.	.	.	.	.	R	.	.	.	E	G
P[23]-Piglet	S	N	A	.	P	S	E	S	.	.	.	V	T	.	I	.	K	E	.	F	T	.	.	.	.	.	.	.	R	.	.	.	E	G
P[23]-Piglet	S	N	A	.	P	S	E	S	.	.	.	V	T	.	I	.	K	E	.	F	T	.	.	.	.	.	.	.	R	.	.	.	E	G
P[23]-Piglet	S	N	A	.	P	S	E	S	.	.	.	V	T	.	I	.	K	E	.	F	T	.	.	.	.	.	.	.	R	.	.	.	E	G
**P[23]-Parent**	**S**	**N**	**A**	**D**	**P**	**S**	**E**	**S**	**Q**	**E**	**N**	**V**	**T**	**T**	**I**	**G**	**K**	**E**	**N**	**Y**	**T**	**T**	**N**	**Y**	**D**	**T**	**T**	**Y**	**R**	**G**	**A**	**R**	**E**	**G**
P[34]-Piglet	K	.	D	.	I	.	T	T	.	S	.	M	.	Q	S	.	L	.	.	Q	.	S	.	.	S	E	.	.	.	.	K	.	L	D
P[23]-Piglet	.	.	.	.	.	.	.	.	.	.	.	.	.	.	.	.	.	.	.	F	.	.	.	.	.	.	.	.	.	.	.	.	.	.
P[23]-Piglet	.	.	.	.	.	.	.	.	.	.	.	.	.	.	.	.	.	.	.	F	.	.	.	.	.	.	.	.	.	.	.	.	.	.
P[23]-Piglet	.	.	.	.	.	.	.	.	.	.	.	.	.	.	.	.	.	.	.	F	.	.	.	.	.	.	.	.	.	.	.	.	.	.
P[23]-Piglet	.	.	.	.	.	.	.	.	.	.	.	.	.	.	.	.	.	.	.	F	.	.	.	.	.	.	.	.	.	.	.	.	.	.

Note: The residues of the parent NPE strains (P[7] and P[23]) are in bold, while dots represent the same residues compared to the NPE strains listed above.

## Data Availability

The data that support the findings of this study are available on request from the corresponding author.

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
