# Peer review of "Role of Pre-Farrow Natural Planned Exposure of Gilts in Shaping the Passive Antibody Response to Rotavirus A in Piglets"

_vaccines, 2023, doi:10.3390/vaccines11121866_

Round 1
Reviewer 1 Report
Comments and Suggestions for Authors
In this report, the authors investigated the effect of the natural planned exposure (NPE) to RVA before farrowing on the passive immune response of piglets. They developed a ELISA-based assay to analyze the level of IgG and IgA in the serum and milk. The results showed that 3-dose administration of pre-farrow NPE resulted in significantly higher IgG and IgA levels in the colostrum and milk, and higher IgG levels in the piglet’s serum. They also indicated that no virus challenging was performed to compare the protection efficacy. While the NPE is used widely in past, it may be a best method. Especially, virus shedding remains high in the gilt and piglet. The administration dose is hard to control to reduce the virus shedding. Moreover, it is necessary to ensured the stock was negative for other pathogens. The authors did not discuss the effect of the virus shedding to environment from a long-term point of view. It may be promote the transmission and reassortant of virus.
1. The authors developed a ELISA-based assay to analyze the level of IgG and IgA in the serum and milk.
But no significant difference in the level of IgG and IgA in the gilt at the different time points. Why? The assay method may have a lack using the expression proteins. It could not analyze the neutralizing antibody level. The total level could not precisely reflect the neutralizing antibody.
2. Anti-porcine IgG and IgA antibody are from? While the author used the serum from gnotobiotic piglets as the negative control. But no results were shown in figure or table.
3. Passive, antibody-based immunity from gilt or sow colostrum and milk is essential to protect piglets from RV infections since in utero transfer of immunoglobulins (Igs) does not occur in swine due to an epitheliochorial placenta. Why are there difference in the serum levels of piglets at day 0 among different groups?
4. In Figure 2, Figures A-C have no scale bar, and the author does not give further explanation on the results of figures A-C.
Author Response
In this report, the authors investigated the effect of the natural planned exposure (NPE) to RVA before farrowing on the passive immune response of piglets. They developed a ELISA-based assay to analyze the level of IgG and IgA in the serum and milk. The results showed that 3-dose administration of pre-farrow NPE resulted in significantly higher IgG and IgA levels in the colostrum and milk, and higher IgG levels in the piglet’s serum. They also indicated that no virus challenging was performed to compare the protection efficacy. While the NPE is used widely in past, it may be a best method. Especially, virus shedding remains high in the gilt and piglet. The administration dose is hard to control to reduce the virus shedding. Moreover, it is necessary to ensured the stock was negative for other pathogens. The authors did not discuss the effect of the virus shedding to environment from a long-term point of view. It may be promote the transmission and reassortant of virus.
Response: We thank the reviewer for the comments. Our results show that low antibody titers generated by rotavirus NPE were sufficient to prevent homologous RVA infection until weaning (day 21) in the absence of virus challenge under natural farm conditions. This study was conducted in a gilt-only commercial farm instead of a controlled environment to assess the effectiveness of NPE in natural conditions. Selecting a gilt-only farm also ensured the most challenging conditions since gilts have been reported to have lower colostrum rotavirus antibody levels than multiparous sows. However, we believe that challenging newborn piglets with RVA would have truly estimated the differences in NPE doses in providing passive protection to the piglets. NPE stock was tested by qPCR to confirm the presence of Rotavirus and absence of other relevant pathogens. NPE material fed to gilts was also sequenced to confirm the presence of different Rotavirus genotypes. NPE protocols are farm specific and strains circulating in a particular farm are included in the NPE material. We agree that high virus shedding by piglets might lead to increased transmission and reassortment events. However, due to complete lack of vaccines against Rotavirus C and availability of only one RVA vaccine, NPE seems to be the only method available to the producers against rotaviruses. Virus shedding results (supplementary table 1) mentioned in this study have been described elsewhere (Anderson et al. 2023, reference 26) and hence were not discussed in detail to avoid repetition.
- The authors developed a ELISA-based assay to analyze the level of IgG and IgA in the serum and milk.
But no significant difference in the level of IgG and IgA in the gilt at the different time points. Why? The assay method may have a lack using the expression proteins. It could not analyze the neutralizing antibody level. The total level could not precisely reflect the neutralizing antibody.
Response: Thank you for the comment. For gilt serum antibody levels (Figure 3), only statistically significant differences among different groups at each time point (at -5W, -3W, F and +3W) were shown in the original submission. However, as pointed out by the reviewer we have now added significance identifiers (asterisks) for significantly different antibody levels for groups at different time points. Unfortunately, we could not perform serum neutralization assay to determine neutralizing antibody titers as Rotavirus A G4 and G5 genotypes used in this study are very difficult to propagate and adapt to the cell culture.
- Anti-porcine IgG and IgA antibody are from? While the author used the serum from gnotobiotic piglets as the negative control. But no results were shown in figure or table.
Response: As suggested, we have now added the catalog numbers and vendor information for anti-porcine IgG and IgA secondary antibodies used for ELISAs (Lines 254-256). Each ELISA plate included a negative (gnotobiotic piglet serum) and a positive control (pooled RVA positive serum) to control plate to plate variation. Same negative and positive control stocks were used throughout the sample testing to maintain uniformity. Negative control well OD values were solely used to calculate the end-point antibody titer of the individual samples (reciprocal of the highest dilution having A410 value greater than twice the mean of negative wells OD). Antibody bar charts only illustrate the geometric mean titers (GMTs) of different study groups calculated using the end-point titers of individual serum or milk samples. However, during sample testing negative control well OD values were always in the range of 0.05-0.08 for IgG and 0.08-0.12 for IgA.
- Passive, antibody-based immunity from gilt or sow colostrum and milk is essential to protect piglets from RV infections since in utero transfer of immunoglobulins (Igs) does not occur in swine due to an epitheliochorial placenta. Why are there difference in the serum levels of piglets at day 0 among different groups?
Response: Day 0 piglet serum samples were collected within 24 hours of birth and colostrum suckling. Since there is no intra uterine transfer of immunoglobulins, differences in day 0 serum antibody levels among different study groups reflect the differences in antibody levels in colostrum fed to the piglets. Factors such as colostrum production by individual sows, individual piglet colostrum intake and litter size might have also resulted in different serum antibody levels in day 0 piglets.
- In Figure 2, Figures A-C have no scale bar, and the author does not give further explanation on the results of figures A-C.
Response: A scale bar is now added to Figure 2 A-C. As suggested, we have added a description of results illustrated in Figure 2 A-C in the results section (line 287-292).
Reviewer 2 Report
Comments and Suggestions for Authors
Maternal antibodies from sows are crucial for protecting piglets from RV infection. The IgA and IgG produced by sows are transported to the mammary gland, mainly through colostrum and milk transfer to piglets. The author studied different NPE administration strategies to detect changes in antibody levels and detoxification patterns in piglets.
However, the authors may strengthen the manuscript by addressing the following points.
1. Is the protein expression level too low when using eukaryotic expression plasmids for the establishment of the Elisa method? Not suitable for clinical production.
2. The format of the table is incorrect.
3. In Figure 2 A-C, there is no ruler.
4. The E1-E4 lane in the figure 2 represents the meaning.
5. Why did detoxification suddenly increase a lot on the 28th day? The detoxification pattern and changes in antibody levels should be further analyzed and discussed.
Author Response
- Is the protein expression level too low when using eukaryotic expression plasmids for the establishment of the Elisa method? Not suitable for clinical production.
Response: We used Expi293 mammalian expression to express G4 and G5 recombinant proteins. Expi293 system is a rapid high-yield protein production system based on high density culture of Expi293F suspension cells. A 200 ml culture of Expi293F yielded approximately 5mg and 3mg of G5 and G4 proteins, respectively. We earlier attempted to use Baculovirus-Insect cell expression system to express these proteins, however the yields were very low (few ug protein from 1L culture) (data not included in the manuscript). Hence, we decided to switch to Expi293 expression system to express G4 and G5 proteins.
- The format of the table is incorrect.
Response: Thank you for the comment. Tables were formatted following the table patterns published in recent papers published in MDPI Vaccines.
- In Figure 2 A-C, there is no ruler.
Response: Thank you for pointing out. A scale has now been added to the images.
- The E1-E4 lane in the figure 2 represents the meaning.
Response: As suggested, we have now included a description of the lanes mentioned in the Figure (Line 310).
- Why did detoxification suddenly increase a lot on the 28th day? The detoxification pattern and changes in antibody levels should be further analyzed and discussed.
Response: The piglets in this study were weaned on day 21 and moved to a different barn (nursery). Sudden increase in RVA fecal shedding on day 28 is primarily due to cessation in supply of protective antibodies particularly IgA from milk (lactogenic immunity) after weaning (day 21). Loss of lactogenic protection post-weaning makes piglets vulnerable to pathogens including rotaviruses. IgA levels in piglet serum started declining rapidly at day 14 and remained very low until day 42. A comparison of piglet serum IgA titers and piglet fecal RVA shedding suggests that possible high environmental load of RVA in the nursery may have been another reason for high fecal shedding in piglets on day 28 considering constant low serum IgA levels in piglets during days 14-42. However, since no environmental samples were collected in this study, it is difficult to make any estimation of RVA loads between environments of the farrowing rooms and nursery. Virus shedding results (supplementary table 1) cited in this study have been described elsewhere (Anderson et al. 2023, reference 26) and hence were not discussed in detail to avoid repetition. However, as suggested a description has been added in the discussion section about antibody levels and RVA fecal shedding in piglets (Lines 542-554).
Reviewer 3 Report
Comments and Suggestions for Authors
Authors described about the antibody response to different dosing regimens of NPE to gilts and transfer to their piglets. To evaluate the effectiveness of RV NPE protocols and differences in antibody response to different proteins, indirect ELISAs were developed to investigate antibody response to RVA G4, G5, P[7] and P[23] genotypes in gilt serum, colostrum/milk, and piglet serum samples. This is a well-written paper containing interesting results which merit publication. For the benefit of the reader, however, a number of points need clarifying and certain statements require further justification. There are given below.
Major comments
L113: Piglets were euthanized 18-24 h post-inoculation… There are no information about methods of sacrifices. Please add methods detail.
L157-161: There are no information about antibodies.
L488: Two doses of NPE in group 1 and one dose of NPE in treatment groups 2 and 3 ?? I think the protocol of this study is different.
L488-486: Do piglets gut closure occur around 36 hours? Please use reference. All figure:
All figure legends are incomplete. Please revise all figure legends. For example, what is asterisk? There are many *, **, ***, ****. But there are no information.
Please add the information and inoculation methods of antigen detail in 2.1. Study design, NPE.
Authors described NPE in this study. But antibody titer declined quickly. Can this method use for pig production? Authors should add more.
Minor comments:
L40: key words are Alphabetical order.
L46: (1) → [1]
L53: [5,6] → [5, 6] Please check through manuscript.
L144-154: There are some plasmid name described about pcDNA3.1+ , pET-24a(+) , pcDNA3.1 (+) . Please use same convention.
L156, L170, L185, L188: Authors use the words“hr”, “hours” ‘min” and “minutes”. Please check through manuscript.
L191: gravity¬-flow ?? Please check this.
L216, L221, L223: 4 times, 4x, 4x →4 times (x), 4x, 4x.
L293: (26) → [26]
L308: at farrowing (F), (F) goes to figure legends.
L343: p23] → p[23]
L480: (32) → [32]
Author Response
Major comments
L113: Piglets were euthanized 18-24 h post-inoculation… There are no information about methods of sacrifices. Please add methods detail.
Response: Piglets were euthanized using carbon dioxide (CO2) gas asphyxiation (please see line 116).
L157-161: There are no information about antibodies.
Response: Antibody details are now included at lines 165-172.
L488: Two doses of NPE in group 1 and one dose of NPE in treatment groups 2 and 3 ?? I think the protocol of this study is different.
Response: The statement in the manuscript refers to the gilt serum levels at 3WPF (-3). To clarify, by 3WPF, gilts in group 1 had received 2 doses of NPE and groups 2 and 3 had only received 1 dose of NPE. The 3rd dose of NPE for group 1 and the 2nd dose of NPE for group 2 was administered at 3WPF.
L488-486: Do piglets gut closure occur around 36 hours? Please use reference.
Response: In piglets, intestinal closure begins approximately 6-12 hours after colostrum feeding and progresses rapidly thereafter to complete closure around 18-36 hours. A reference (Weström et al. 1984) has now been added.
All figure legends are incomplete. Please revise all figure legends. For example, what is asterisk? There are many *, **, ***, ****. But there are no information.
Response: As suggested all figure legends have been revised and details of sub-figures added. Asterisks indicating significant difference among groups at different time points have been defined in Figure 3, 4 and 5.
Please add the information and inoculation methods of antigen detail in 2.1. Study design, NPE.
Response: We have now added the details of NPE inoculation to the gilts at lines 127-129.
Authors described NPE in this study. But antibody titer declined quickly. Can this method use for pig production? Authors should add more.
Response: At present, many commercial swine farms in the US use pre-farrow Rotavirus NPE protocols to prevent Rotavirus A (RVA) and C (RVC) infection in nursing piglets. One of the main reasons of using NPE methods is the lack of modified live virus vaccine (MLV) against RVC due to its inability to propagate in the cell culture. Hence, NPE is the only method available to prevent RVC infection in neonatal piglets. There is only one commercial swine RVA vaccine available in the US (Merck Animal Health) and it does not provide cross protection against RVB and RVC. Our data suggests that colostrum/milk IgG levels reached baseline starting at day 7. However, colostrum IgA levels after declining at day 7 maintained a steady increase until day 21 (weaning), which likely reflects the increased number of RVA-specific IgA plasmablasts in the mammary glands and continuous supply of secretory-IgA in the colostrum/milk. Serum IgA levels are a good indicator of intestinal IgA content and high pathogen-specific IgA levels in milk have been associated with lower incidence of RVs in swine. Overall, piglet serum IgA levels and RVA fecal shedding data (supplementary table 1) suggest that low antibody titers in piglets’ serum were sufficient to prevent homologous RVA infection in piglets until weaning (day 21). We have added a sentence in the conclusion section (lines 619-621) to highlight the importance of antibody levels against RVA.
Minor comments:
L40: key words are Alphabetical order.
Response: Key words are now listed in alphabetical order (Line 40).
L46: (1) → [1]
Response: Correction made.
L53: [5,6] → [5, 6] Please check through manuscript.
Response: Suggested change made throughout the manuscript.
L144-154: There are some plasmid name described about pcDNA3.1+ , pET-24a(+) , pcDNA3.1 (+) . Please use same convention.
Response: Thank you for pointing out the mistake. All plasmid names now follow the uniform convention throughout the manuscript.
L156, L170, L185, L188: Authors use the words“hr”, “hours” ‘min” and “minutes”. Please check through manuscript.
Response: Thank you for pointing out. All time frames have been changed to either “hrs” or “min” throughout the manuscript.
L191: gravity¬-flow ?? Please check this.
Response: Corrected (Line 202)
L216, L221, L223: 4 times, 4x, 4x →4 times (x), 4x, 4x.
Response: Corrected (Lines 235 and 238).
L293: (26) → [26]
Response: Corrected (Line 316)
L308: at farrowing (F), (F) goes to figure legends.
Response: Thank you for pointing out. We deleted symbol (F) for farrowing at line 331 as figure 3 legend describes all X and Y axis values.
L343: p23] → p[23]
Response: Corrected (Line 370)
L480: (32) → [32]
Response: Corrected (Line 520)
Reviewer 4 Report
Comments and Suggestions for Authors
Review letter
Dear Editor,
Thank you so much for your invitation for reviewing manuscript finished by Dr Kumar et al entitled “Role of pre-farrow natural planned exposure to gilts in shaping the passive antibody response to rotavirus A in piglets”. In the study, authors aim to solve low efficacy of Natural planned exposure (NPE) of rotavirus A and relevant passive antibody response. To their aim, authors conducted a longitudinal study to investigate the dynamics of genotype-specific antibody response to different doses (3, 2 and 1) of Rotavirus A (RVA) NPE (genotypes G4, G5, P[7] and P[23]) in gilts and transfer of lactogenic immunity to their piglets. Authors found that in day 0 colostrum samples, group 1 had significantly higher IgG titers compared to the control group for all four antigens, and either significantly or numerically higher IgG titers than group 2 and 3. They also found that group 1 also had significantly higher colostrum IgA levels than the control group for all antigens (except G4), and either significantly or numerically higher IgA levels compared to group 2 and 3. Moreover, authors found that, in piglet serum, group 1 piglets had higher IgG titers for all four antigens at day 0 than other groups. Moreover, authors found that RVA NPE stimulated antibodies in all groups regardless of the treatment doses and prevented G4, G5, P[7] and P[23] RVA fecal shedding prior to weaning in piglets. They also found that the G11 and P[34] strains from pre-weaning piglets differed at multiple amino acid positions with parent NPE strains. Eventually, authors concluded that results of this study suggest that group 1 NPE regimen (3 doses of NPE) resulted in the highest anti-RVA antibody (IgG and IgA) levels in the colostrum/milk, and highest IgG levels in piglet serum.
In general, the study is well designed, and the writing of the manuscript is very well. I have some comments to authors:
Q1: for materials and methods, could authors add more information such as producer location, catalog number and so on.
Q2: Figure 2 A, B, and C, please add scales;
Q3: For Figure 3, please add detail explanations for each sub-figures for Figure legends;
Q4: Same to figure 3, for Figure 4, please add detail explanations for each sub-figures for Figure legends;
Q5: for Figure 5, please add detail explanations for each sub-figures for Figure legends;
Q6: For bar figures, I saw the variations seem to be large, which variation you used, mean ± SD or mean ± SEM?
Comments on the Quality of English Language
Minor revision is needed, since I found some grammar errors.
Author Response
In general, the study is well designed, and the writing of the manuscript is very well.
Response: Thank you very much for the words of encouragement. Please find our response to the queries below.
Q1: for materials and methods, could authors add more information such as producer location, catalog number and so on.
Response: As suggested, catalog numbers have been added in the material and methods section. We apologize, swine farm location cannot be revealed in the manuscript as it is protected under client confidentiality agreement.
Q2: Figure 2 A, B, and C, please add scales;
Response: In Figure 2, images A, B and C were taken using a cell phone camera with microscope set at 10x magnification. Accurate scale bar information is not available now and hence magnification (10x) has been added to the images instead of a scale bar.
Q3: For Figure 3, please add detail explanations for each sub-figures for Figure legends;
Response: As suggested, we have included relevant explanations for sub-figures of Figure 3, 4 and 5 in the figure legends. Identifiers for p-values have also been added.
Q4: Same to figure 3, for Figure 4, please add detail explanations for each sub-figures for Figure legends;
Response: Please see response to Q3.
Q5: for Figure 5, please add detail explanations for each sub-figures for Figure legends;
Response: Please see response to Q3.
Q6: For bar figures, I saw the variations seem to be large, which variation you used, mean ± SD or mean ± SEM?
Response: All antibody titers are represented as Geometric mean titers (GMTs) with confidence interval. High variation seen in gilt serum and colostrum/milk antibody titer bar charts is due to low number of samples in each group (approximately 12 animals/group) and broad range of dilutions used to determine end point titers (1:200, 1:400, 1:800, 1:1600, 1:3200, 1:6400, 1:12800 and 1:25600). Any serum/milk sample that appeared to cross the highest dilution (1:25600) on ELISA testing was retested with more dilutions to accurately determine the end-point titer. In contrast, variation is low in piglet serum titers because of the large sample size (approximately 60) at each time-point.
Round 2
Reviewer 1 Report
Comments and Suggestions for Authors
The manuscript has been much improved.
Comments on the Quality of English LanguageThe manuscript needs minor editing of English language.
Reviewer 3 Report
Comments and Suggestions for Authors
There are no comments.
Reviewer 4 Report
Comments and Suggestions for Authors
Thank authors for their revision about the manuscript.